# Empowering Physically Disabled People in Vietnam: A Successful Microenterprise Model

June Alexander [1,*], Claire Hutchinson [2] and Greg Carey [2,3]

1   Disability and Community Inclusion Unit, College of Nursing and Health Sciences, Flinders University, Adelaide, SA 5042, Australia
2   Caring Futures Institute, College of Nursing and Health Sciences, Flinders University, Adelaide, SA 5042, Australia; claire.hutchinson@flinders.edu.au (C.H.); drgregcarey@gmail.com (G.C.)
3   College of Education, Psychology and Social Work, Flinders University, Adelaide, SA 5042, Australia
*   Correspondence: june.alexander@flinders.edu.au; Tel.: +61-8-8201-3902

**Abstract:** *Background:* Disabled people in Vietnam are some of the most vulnerable to disadvantage. Employment involving microenterprises can provide economic empowerment and wealth generation. This qualitative study aims to address a gap in the literature regarding the establishment of microenterprises for physically disabled people in Vietnam. *Method:* Semi-structured interviews were undertaken with seven physically disabled individuals, including the founder and Director of 'Company of Grace' a non-governmental organisation with a mission to support physically disabled people in establishing their own microenterprises. Company of Grace (COG) supported six entrepreneurs in establishing microenterprises that provided English language instruction to school-aged children after regular school hours. Data were analysed utilising a framework that improves the probability of entrepreneurial success in developing countries. This framework aided in examining the approach of establishing microenterprises by the non-government organisation. *Results:* The physically disabled entrepreneurs reported earnings above average wages and feeling empowered by participating in the microenterprises. These feelings of empowerment were reportedly associated with greater independence, increased self-efficacy and confidence in planning for their futures. *Conclusion:* Microenterprises, exemplified by COG's model, empowered disabled individuals to teach English, enhanced student engagement and fostered confidence and economic self-sufficiency among disabled entrepreneurs, thereby making a notable contribution to entrepreneurship for disability inclusion.

**Keywords:** physical disability; employment; microenterprise; Vietnam





## 1. Introduction

Vietnam has gradually developed from one of the world's poorest countries into a lower-to-middle-income country over the last 25 years [1]. This change in status is due mainly to Vietnam adopting economic reforms such as the encouragement of the private sector [2]. Nevertheless, an estimated one in five individuals still reside below the poverty threshold [3]. Some of the people most at risk of disadvantage are disabled people, with disability being significantly correlated with poverty [4]. This study adopts the World Health Organization's social model perspective on health and functioning, as outlined in the International Classification of Functioning, Disability, and Health [5]. Adopting this model offers a focus on social structures and other barriers to disabled people's inclusion, such as societal attitudes and resource accessibility, and how these elements shape the experiences and outcomes of disabled individuals in the context of microenterprises.

The only large-scale survey on disabled people in Vietnam highlighted almost half (49.7%) of disabled individuals had physical disabilities [6] and disabled people were three times more likely to be unemployed compared to non-disabled people (14% vs. 4.3%) [7], with most disabled people not being able to cover their daily expenses [7,8].

A key to overcoming poverty is employment [9] which creates independence and self-sufficiency [10]. However, for physically disabled people in Vietnam, it can be difficult to obtain and maintain employment [11]. Whilst physically disabled people encounter barriers to employment that are complex and unique to themselves as individuals, there are common shared factors that can be identified. For example, disabled people often have limited opportunities to access education resulting in limited skills making it difficult for them to compete for jobs [12,13]. It is reported broadly that disabled people also face many barriers to employment, such as discrimination from co-workers [14]; prejudice from employers [15]; and societal barriers, i.e., inaccessible transport, public buildings, and accommodations [16]. Personal influences can also act as barriers to employment, such as self-esteem, confidence, own actions and initiative, knowledge and experience, past influences, and personality-related factors [17].

Addressing barriers to employment and thereby reducing the number of disabled people living in poverty requires significant policy change [14]. In particular, the literature highlights the opportunities presented by entrepreneurship for disabled people. Self-employment can provide disabled people with independence and allow individuals to set their own schedule which might be more sustainable than working to fixed employment hours. It has been found in developing countries that disabled people are more likely to be self-employed than non-disabled people [6]. Self-employment is referred to as employment performed for personal profit instead of for wages and can be used as a viable employment outcome for disabled people [18]. One such model of self-employment is a microenterprise. Microenterprises are small businesses, often established with limited resources and operated by individuals or small groups, that provide goods or services in the local community [19]. Globally, microenterprises have proven to be one of the most significant means through which people on low incomes can escape poverty [12,20–22], offering a pathway through economic empowerment and the creation of wealth. Furthermore, microenterprises for disabled people have been shown to provide a range of personal outcomes, such as independence and enhanced confidence and self-esteem [23,24]. However, Lingelbach, De La Vina and Asel (2005) [25] argue that policy approaches alone to entrepreneurship development, particularly in developing countries, may not be beneficial. They propose three characteristics—opportunity, financial resources, and apprenticeship/human resources—that offer a framework for entrepreneurial success in developing countries [25].

### 1.1. Challenges for Disabled People in Vietnam

Disabled people in Vietnam continue to be marginalised in society, often facing cultural barriers [26]. Like many Asian neighbours, Vietnam tends to perceive disability as a mystical punishment for sins committed by one's parents or ancestors, and disability often brings shame and guilt to the family [27]. In 2010, the 'Law on People with Disabilities' was enacted as the initial document safeguarding the rights of disabled individuals. This marked a notable change in Vietnam's approach toward disabled people [28]. Despite this legislation, disabled people within Vietnam still tend to be negatively perceived [29]. For example, employers and the wider community carry negative attitudes towards disabled people and consider them as objects for charity rather than being valued for who they are [30]. The misguided prejudice amongst employers means that even people with a mild form of disability have difficulty accessing education and employment [26]. Discrimination can also involve a failure to provide reasonable workplace accommodations, such as adjustments to employment obligations, modifications, or policies for disabled individuals. Although Vietnamese law stipulates that employers must ensure reasonable adjustments, there may be a gap between policy and implementation [31].

Limited opportunities for participation in employment can lead to profound effects on the individual's quality of life. There has been much international evidence demonstrating that participation in employment can improve quality of life. Examples include significant

increases in confidence in making personal decisions and elevated feelings of self-worth and empowerment in daily life [31].

Vietnam has developed several approaches to address employment for disabled people. For example, the National Coordinating Council on Disability (NCCD) coordinates employment and training programs for disabled people. However, the provision of employment services is still an issue [32] with many enterprises engaging in production with low profit margins [8]. As such, the opportunities available for disabled people seeking fulfilling and sustaining employment appear little different from other developing nations [27].

The Vietnamese Government encourages companies to recruit disabled workers by setting an employment quota for each industry area. However, many employers continue to choose not to employ disabled people [30], with quota-levy systems generally accepted as ineffective in addressing the employment gap for disabled people [32].

In another initiative, the Government, NGOs and private organisations have established some 599 Vocational Training Schools for disabled people [27]. These schools provide training in a range of industries, including sewing, hairdressing, silk printing, electronics and electronic repair, embroidery, signwriting/advertising, motorbike mechanics, graphic design (computer-based), accounting and basic office computing. Training tends to be of 6–12 months duration, and though exact data are not available, it is estimated that 50 percent of training participants are able to find jobs after completing their vocational training course [7,33]. This outcome raises concern about the suitability and effectiveness of the programs for the remaining training participants. Furthermore, it is reported that as many as an estimated 50 percent of disabled people would like to start a home-based business after training [27]. A home-based business is an enterprise or commercial venture that is operated and managed from a residential location, typically the business owner's home. Home-based businesses are favoured by disabled individuals, due to challenges encountered when seeking traditional employment opportunities [27].

Since 2005, the Vietnamese Government has provided disabled people with either a job placement or job counselling for employment opportunities and a service introducing disabled people to prospective employers. However, very few people have accessed these services primarily due to the small number of centres and a lack of skilled staff available to give well-considered and informed support in a timely manner [34]. In addition, there are many physical and attitudinal barriers experienced by disabled people in accessing these centres and education in general [35]. The Vietnamese cultural emphasis on care and protection results in disabled individuals receiving essential care, food, and shelter. However, societal inclusion, particularly in workplaces, remains limited due to the prevalent belief that disabled people are perceived as incapable [27].

Lastly, the Vietnamese Government provides support through a system of low-interest loans allowing access to start-up capital for people who wish to be either self-employed or to establish their own microbusiness [6]. These loans, along with several other credit programmes established by local and international NGOs, can be applied for but are highly competitive, with disabled people reportedly having difficulties securing credit [8].

Faith-based organisations (FBOs) are widely acknowledged for their roles as contributors to addressing social justice issues and the economic advancement of people in developing countries [36]. In Vietnam in 2005, there were an estimated 100,000–200,000 community-based organisations, including FBOs; however, there is little information available about these groups because of their informality [27]. Many start-up organisations and individuals, including disabled people, may seek start-up funding from an FBO which is traditionally supportive of people who would ordinarily be unable to access the usual lines of credit [37]. This study seeks to outline the process by which microenterprises in Vietnam that provide English language instruction are developed by physically disabled entrepreneurs, supported by an Australian faith-based organisation (FBO) and international donations.

### 1.2. Company of Grace (COG)

COG is an NGO and a not-for-profit entity, receiving funds and in-kind support from an Australian FBO and other international FBOs. COG was established by the Director, Hoa (Peter) Stone, in 2011. COG has a strong empowerment and self-sustainability focus and operates in one of Ho Chi Minh City's 24 districts.

The history of COG developing microenterprises for physically disabled people has been relatively short and, as COG's Director admits, was originally developed through much trial and error. COG's Director has no qualifications or formal education in either business or disability; he does, however, have lived experience of disability which has driven him to become a 'disability change agent' [38]. COG's mission is to provide educational and vocational training services to disabled people in Vietnamese communities, to empower them to live independent and self-sustaining lives. Future goals include to be 100% self-sustainable and 100% managed by disabled staff. COG is operated by a board consisting of nine volunteers with a variety of skills and experiences, including pastors, accountants, lawyers, and disabled people. The organisation has purchased land and buildings which provide a residence for physically disabled people and staff, as well as five classrooms and an enclosed playground area. The classrooms are used by physically disabled entrepreneurs to provide English classes to the local community.

On opening in 2016, 150 students attended the English classes at COG, and the numbers have been growing steadily since. The English classes are run from 4.30 p.m. to 8.30 p.m. Monday to Saturday. The classes are taught by seven physically disabled people, four women and three men, who have learnt English as a second language. There are 24 classes with a variety of levels and session times with class sizes varying from 7 to 24 students for year levels 3–8 (early primary/elementary school to the first year of high school). At the time of this study, COG provided English classes for around 200 local children.

Due to its emergence as a global language [39], English is highly valued in Vietnam as the language enabling growth in many areas including scientific and commercial endeavours [40]. English is the main foreign language taught and used in Vietnam, and its use in the general community has developed with unprecedented speed [41]. Ho Chi Minh City has the largest number of teachers and learners of English in the country and many language centres in universities, high schools, government agencies and private enterprises [41]. Since 2011, English has been a compulsory subject in primary schools with four 40 min periods per week starting from Grade 3 incorporated into the curriculum [42]. With a desire to promote the teaching and learning of English, the Ministry of Education and Training (MOET) introduced systems to ensure the quality of teaching English is maintained [41]. However, low scores on final high school results in English have been reported and could be attributed to the current teaching methodologies [43]. Much of the difficulty in learning English has been attributed to a mismatch between recognised effective teaching pedagogy and traditional Vietnamese classroom cultures. Rote learning is the preferred approach in most Vietnamese Government schools, where students are encouraged to memorise words and grammar rules by heart, often at the expense of comprehension, which is an important part of the learning process [44]. Teaching practice involves *'learner involvement, allowing learners' choice, changing teachers' and students' roles, and breaking down hierarchic barriers in the classroom challenges basic Vietnamese socio-cultural and educational values'* [42] (p. 2). The point of difference for COG's English teaching is the pedagogy of learner engagement. In part, this can be traced to modifications and adaptations that the disabled teachers require to teach effectively.

### 1.3. The Current Study

This study examines how one NGO in Vietnam (COG) assisted in establishing microenterprises for physically disabled people, highlighting the practical interventions that supported physically disabled entrepreneurs to develop their own microenterprises teaching English classes in a developing country. Responding to the call for an increased

understanding of employment for disabled people [31], this study investigates the approach and outcomes of COG using Lingelbach, De La Vina and Asel (2005) [25] three attributes of entrepreneurship in developing countries that contribute to success and growth as an analytical lens.

## 2. Materials and Methods

This qualitative study sought to describe COG's development of microenterprises for physically disabled people in Vietnam. Lingelbach, De La Vina and Asel's (2005) [25] characteristics of entrepreneurial success in developing countries were utilised to guide data analysis [45]. Lingelback's framework provides an outline of characteristics of entrepreneurship in developing countries that improve the probability of success, and this study aimed to outline the contributing factors that may lead to the success of disabled people operating their own microenterprises.

This approach was selected to provide a foundation for follow-up work which may be valuable in the design and implementation of similar projects in similar political and economic environments [46]. The methodology allows an insight into the evolution of one NGO and its partnership with an FBO to empower physically disabled people through meaningful employment.

Ethics approval for this study was provided by Flinders University Human Ethics Committee (approval number 7290).

### 2.1. Setting

This study involved one organisation (COG) located in Ward Hiep Binh Phuoc, District Thu Duc, Ho Chi Minh City, South Vietnam. Twelve interviews in total were conducted over a period of nine days.

### 2.2. Participants

In total, 12 people were interviewed for this study, including 3 parents of children who attended the English classes, 2 Australian volunteers and 7 physically disabled people. Only the seven voices of disabled people are included and reported in this article. The disabled people included the Director and founder of COG and six teachers (see Table 1). The teachers were four females and two males, aged 19–33 years old (average age 26 years). Physically disabled participants had disabilities that impacted their arm, hand, and leg movements. Not all participants reported having a diagnosis that was medically identified. While access to healthcare in Vietnam for disabled people is reportedly good, there are extra costs associated with examination and treatment [6]; hence, a diagnosis may not be sought. Teacher participants self-reported the causes of their disabilities as congenital (2), stroke, spinal meningitis, polio and high fever. Most participants reported developing the disability from a young age, although one reported acquiring her disability in her early twenties as a result of a stroke.

The education levels of the six teacher participants were as follows: tertiary studies (2), Year 6, Year 8 and Year 9 (students in Vietnam in Year 6 are aged 11–12 years), and one participant reported they did not attend school. Five out of six participants considered they had no English education prior to attending COG, however, one teacher had studied English at the university level.

Four out of six teacher participants had not previously been employed before coming to COG. One participant sold clothing apparel online but made little money and relied on the support of her parents to live; another had been a kindergarten teacher until she had a stroke which led to her disability. She then lost her job because she was perceived as 'diseased'. The previous experience of the other participants without previous jobs was varied; one participant had been a university student but was forced to abandon his studies in the second year because he could not write quickly enough. The other participants had attended a government vocational school for disabled people but reported that the skills learned in the school were not useful in obtaining employment. The experiences described

exemplify how societal attitudes and structural barriers, rather than inherent disabilities, significantly impact disabled individuals.

**Table 1.** Demographics of participants with disability.

| ID | Position | Sex | Age | Highest Education Level | English Prior to Attending COG | Employed Prior to Company of Grace | Reported Disability | Age of Onset of Disability |
|---|---|---|---|---|---|---|---|---|
| #1 | Teacher | F | 33 | Tertiary Education | No | Yes, but had lost her job after her stroke | Paralysed after stroke | Early 20s |
| #3 | Director | M | 49 | High School in Australia | Yes | Yes | Polio | Child |
| #5 | Teacher | M | 28 | Year 8 | No | No | Spiral meningitis | 12 years |
| #7 | Teacher | M | 33 | Some Tertiary Education | Yes | No | Polio | Congenital |
| #8 | Teacher | F | 19 | Year 9 | No | Yes, own online business but did not make enough money | Physical disability affecting arms and legs | Congenital |
| #9 | Teacher | F | 24 | Year 6 | No | No | Paralysed right side | 12 years |
| #10 | Teacher | F | 24 | No formal schooling | No | No | Weak arms and legs | 5 years |

### 2.3. Semi-Structured Interviews

A semi-structured interview guide was developed to structure the interviews. Interview questions were developed by addressing the goals and objectives of the study and reviewing relevant literature. The Director was asked a series of questions related to the history and evolvement of COG microenterprises. Participants provided demographic information, previous employment experience, their involvement in teaching English at COG and the impact owning microenterprises had on their lives. The Director's interview was 100 min while other interviews varied from 15 min to 36 min.

### 2.4. Procedure

A local translator was employed to be present during the interviews and interpret when necessary. Participants who had English as a second language expressed gratitude for having an opportunity to practice their English skills. The translator was primarily utilised to clarify word meanings or to ensure correct understanding for both the researcher and study participants. All participants gave written consent to participate in the study, and the study's information forms and consent forms were available in both English and Vietnamese. All participants were offered confidentiality; however, COG's Director requested his own name and that of the organisation be reported. All other participants were made aware of this request and agreed to be part of the study knowing their names would not be divulged but total anonymity may not be possible. Accordingly, an ethics modification was obtained.

### 2.5. Data Analysis

Data were analysed using content analysis and were theory-guided using the predetermined codes of (1) opportunity (2) financial resources and (3) apprenticeship/human resources [25] applied to the qualitative interview data [25,47]. Merging content analysis with theory-guided codes allowed the exploration of data within a framework of existing theory, while also being open to new insights that emerged during the analysis process,

providing a structured and theoretically informed approach to qualitative data analysis. The findings both supported and extended the theoretical framework.

Qualitative content analysis has two methodological approaches (openness and theory-guided investigation), but this dichotomous approach can be adapted by enabling existing categories to be modified or new categories developed during analysis [47,48]. This was important as one extra category, 'empowerment', was added to Lingelbach, De La Vina and Asel's (2005) [25] framework to reflect the findings.

## 3. Results

### 3.1. Director

According to COG's Founder and Director, the path to developing opportunities for the establishment of microenterprises for disabled people was not linear. Initially, the intention of COG was to be a service for disabled children. However, a chance meeting with Participant #7 led COG's Director to change the service's focus from children to providing an employment pathway for physically disabled adults. COG's Director recognised that the current employment services for disabled people were not always meeting individuals' needs. Donations mainly from an FBO in Australia were used to buy land and renovate existing buildings and build new ones, at an initial cost of AUD 30,000 in 2011.

In the early days of COG, four physically disabled men were provided with a stipend, accommodation and meals. These men utilised existing skills to generate an income by fixing computers, using photo editing programs to make stationery and selling oil paintings. Over time, the group grew to also include four women; they initially made jewellery, crafts and wooden items. However, it was soon evident that people were only buying the goods and services out of charity. Hoa reported, *'Items not as good . . . people were buying out of charity . . . sold [for] less than abled-bodied items.'* Furthermore, the impact of the group's disabilities presented barriers to the supply of these goods and services. For example, an individual's disability sometimes did not allow consistent quality production and/or their disability impacted the time to produce the item which could not be recouped in the sale price. These factors indicated that these existing enterprises were not going to lead to consistent substantial wages for the disabled individuals.

The group was encouraged to experiment with any ideas they had for microenterprises. Hoa commented, *'It was [participant #7] idea to teach English. Other businesses died but English classes [were] successful.'* This experimental phase was only possible because of the monthly stipend and small amounts of start-up capital provided by the Australian FBO through COG. Equipment such as televisions, computers, desks, chairs and air conditioners and internet access were purchased. While COG financially supported the initial setup, monies earnt by teaching English were retained by each of the microenterprise owners. Class sizes continued to grow quickly through word of mouth in the local community. In fact, Hoa joked, *'In the early days they [COG Board] asked, 'What in the heck are you doing right to have 150 kids here?'*' Competition to provide English classes is strong in the district, but COG's Director attributed the popularity of the COG English classes to the following:

(1) The use of PowerPoint presentations. Initially, PowerPoint presentations were utilised because disabled teachers found it difficult to write on a black/whiteboard. COG then developed these into interactive PowerPoint presentations to engage students. PowerPoint presentations are not generally available in schools in Vietnam.

(2) The relaxed, interactive atmosphere of the English classes. Regular school classrooms in Vietnam are quiet and orderly, with little teacher–student interaction.

(3) Children coming to COG after school where they could play on the COG playground and purchase drinks and snacks.

The English classes are based on the standardised books and curriculum used in the local school system, providing English tutoring that enhances lessons at school. COG acquired the necessary licences for both a not-for-profit organisation and to teach English and is subject to audits every one to two years. There are plans for the disabled teachers to

become qualified to teach English to an international Teachers of English to Speakers of Other Languages (TESOL) standard [49].

COG's Director acknowledged that not every disabled person will be able to teach English, providing the example of one COG participant's poor muscle control limiting his ability to speak clearly enough to teach effectively, for whom other activities were being explored. It was reported that it had taken three years to teach the physically disabled teachers English to their current standard. Given some teachers had no schooling or only a primary school education, this could be seen as quite an achievement. All teachers learnt English principally from the disabled teacher who studied English at a university level. Hoa reported, *'[Participant #7] had English grammar knowledge but others had no English . . . they learnt from volunteers, me and [participant #7].'* The teachers reported learning had been further enriched by learning pronunciation from internet sources such as Google and conversational training with the many overseas volunteers and visitors.

When the disabled people first came to COG, they received instruction in life skills including problem-solving, money management and cooking, as well as lessons designed to build self-confidence. *'People [with disability] come to teaching [learn] life skills on a weekend . . .problem-solving, money, encouragement, what make you happy, talk to each other. They liked being part of a community'* (Hoa). Entrepreneurial skills were practiced by the buying of goods such as greeting cards and jewellery from local producers and selling them for a small profit. The aim was for the disabled participants to be the 'middleperson' in business and learn valuable business skills. The teachers also sold drinks and snacks before English classes and received the profits from the sale of goods as well as an income from the classes they taught. The sale of snacks also provided an opportunity to develop entrepreneurial skills.

COG's Director recounted previously employing non-disabled people to assist with the microenterprise's development but recognised an immediate power imbalance. He reported that some *'pushed too hard'* and did not recognise the needs of physically disabled people. COG's Director recognised that he *'needed to pull the power [from the non-disabled]'* because people without a disability were *'making decisions for people with disability'*. COG has since introduced a policy of not employing non-disabled staff and allowing non-disabled people to be volunteers only.

### 3.2. Physically Disabled Teachers

Physically disabled teachers reported some referrals to COG came from government agencies, but most were through word of mouth in the local community. Before working at COG, all teachers reported either not having jobs or having low-paying or low-status jobs. For example, Participant #7 reported earning only enough to get himself to work, *'Paid for only fuel. . . Not enough for living'*. Participant #10 also reported earning very little money, stating that *'Sewing was hard and slow'* and *'[I was] just studying, not a job'*.

The teacher participants, similar to the Director, reported the early attempts at microenterprises as unsuccessful due to their physical disability leading to a lack of consistency in producing high-quality goods and services, and therefore not creating a sustainable wage. However, the use of adaptations (computers, special keyboards, software and access to the internet) allowed the teachers to deliver popular and interactive English classes from which the teachers reported an income of VND 2,000,000 a month. Further income from the sale of souvenirs (such as cards and jewellery made by others) and income from the sale of snacks and drinks before and after English classes were reported as 680,000 VND a month. The teachers reported their income was relatively high in comparison to the average wage and well above what most disabled people would expect to earn in Vietnam.

All teacher participants reported having grown in confidence since coming to COG, stating that previously they could not look people in the eyes or were *'disappointed and shy'* (Participant #9). Participant #5 stated, *'[f]or a long time I have [had] no joy'*, in which he was referring to the embarrassment he felt that he had brought to his family by his disability, a feeling which eventually led to him leaving his family home. After involvement with COG,

many disabled teachers reported having dreams of being able to travel, marry and have their own children, thoughts they previously would not have dared to consider. Participant #9 elaborated further:

> *I dream I can go overseas and learn some more. Coming to Company of Grace has helped me with my dreams. Company of Grace has bought a man to me [in reference to her new relationship]. With love, new dream.*

While all teachers reported enjoying teaching English, some had plans to open different types of microenterprises such as selling souvenirs, a bookstore for students and a fashion shop. However, these teachers also intended to keep working as English teachers to enable them to save enough money to commence new enterprises sometime in the future. Participant #10 said, *'I want to open [a] shop for shampoo and souvenirs. Saving for that'*.

### 3.3. Future Aspirations

Teacher participants reported future aspirations that included travel, marriage, having a family and starting other microenterprises, all of which they felt were now possible due to the establishment of the English class microenterprises. In fact, four female disabled teachers had moved out of COG accommodation into their own shared rented premises and were planning to begin new microenterprises—teaching English classes separate from COG's premises. COG will continue to provide the necessary support, including business advice and, if needed, food and accommodation if the new microenterprises fail to thrive in the new location.

## 4. Discussion

COG's achievements in establishing microenterprises for physically disabled people were multi-faceted. The analysis of the interviews was coded into the three components of entrepreneurial development: (1) opportunity (2) financial resources and (3) apprenticeship/human resources [25]. Another component, 'empowerment', was also identified and discussed.

### 4.1. Opportunity

'Opportunity-based entrepreneurship' that bases a start-up business on an opportunity in the market as opposed to 'necessity entrepreneurship' where a business begins because there is no other means of generating income is preferable because it allows for more growth [25]. Early in the participants' journey of becoming entrepreneurs, there were initial attempts to utilise 'necessity entrepreneurship' when they made and sold a variety of items such as stationery, jewellery and crafts. However, ultimately, this was not successful. Eventually, English-teaching microenterprises were established, building upon opportunity by addressing a gap in the local market to provide interactive English classes for school-aged children. Using opportunity-based entrepreneurship is important but an often-missed supporting success factor of businesses [25]. While the COG Director admitted to not having formal education about business development, the microenterprises ultimately thrived because of their ability to recognise and build upon local 'opportunity.'

### 4.2. Financial Resources

Limited bank lending and limited personal savings result in inadequate access to start-up capital in developing countries [8,25]. Furthermore, credit alone is not the only concern when establishing a business. In a previous model of microenterprises in developing countries, i.e., the Bangladesh program, Income Generation for Vulnerable Group Development Program (IGVGD), it was discovered that food rations and training need to continue for more than 2 years to support the establishment of a microenterprise [50]. Programs must address basic survival needs such as food, housing, education, transportation, energy, health and safety needs. COG initially provided a wage, food and accommodation for disabled people while they were experimenting with what microenterprise may work for them. Training provided included cooking skills and confidence building. The training

was provided for four years, and this support will continue until the teachers feel it is no longer required.

Start-Up Capital

For any new business venture in any country, gaining loans to start the business and obtain necessary equipment and supplies can be difficult. Vietnam introduced support for small and medium enterprises [51]; however, there are limitations in the process of borrowing capital [50]. Traditional sources of credit, such as banks and investment groups, prefer business entrepreneurs with a proven record of success and households with stable incomes and multiple earning sources [52,53]. The borrower's character, business prospects and ability to repay are scrutinised and assessed during the application process. Unfortunately, most disabled people in Vietnam have insufficient personal savings and do not qualify for loans [4]. The availability of loans at reduced interest rates is vital for disabled people if they are to succeed as entrepreneurs [54]. Until that becomes common practice, individuals and organisations with creative ideas rely on local, community or international grants to help establish their microenterprises [39]. International donations and in-kind support through an Australian FBO provided the funds for the establishment of the COG program and associated equipment such as buildings, computers and televisions.

COG was able to obtain initial funds through the networks of the Director. COG's Director built partnerships with individuals and FBOs in several countries who have provided financial aid and practical support. Using donations, COG developed an innovative employment approach that enabled physically disabled people to reach their aim of starting their own microenterprises. The priority for COG was to develop a structure that allowed employees to build personal savings, so they did not require a loan or grant when starting a microbusiness. Each participant earnt their own money depending on how many hours they worked but were utilising COG's facilities and equipment. The plan was that, once confident enough, they would start up microenterprises in different locations and different types of microenterprises. To be successful, the approach required developing opportunities and providing appropriate support and training so that individuals produced enough income to provide a sustainable wage and build savings for their own future initiatives.

The money earned from teaching classes at COG was reported as over VND 2,000,000 per month. The monthly average wage in Vietnam was VND 7000 in 2023 [55], and people who are highly skilled earn VND 12,000,000 per month [56]. The significant difference between actual monthly earnings and Vietnam's average monthly earnings meant that teachers had the potential to develop substantial savings each month. The teachers interviewed reported that their income from teaching English allowed them to earn sufficient income to live and establish savings whilst also learning valuable skills. Highlighting the importance of earning a sufficient income, Fields [52] (p. 1) states that *'poverty is often not primarily a problem of unemployment, but a problem of low labo[u]r market earnings among the employed.'* COG's model supplied the initial resources required for the English classes, including buildings, computers and televisions. This initial support from COG allowed disabled entrepreneurs to secure savings so they would not be reliant on an often difficult-to-obtain loan. The contribution from the FBO had an additional advantage as it took the form of a donation, and as such, there was no expectation for it to be repaid, providing a solid financial foundation for the microenterprises.

### 4.3. Apprenticeship and Human Resources

Entrepreneurship is made more difficult without the benefit of mentorship and apprenticeship [25]. Entrepreneurial foundation structure includes economic, financial, technological and business literacy, as well as entrepreneurship training, mentors and role models [22].

### 4.3.1. Entrepreneurship Training

While the charisma and networks of a leader may be able to garner initial funding, an enterprise needs to be viable beyond the start-up phase. Entrepreneurs need to progress to technical proficiency, moving beyond low capital-demanding activities if they are to succeed [53]. Unfortunately, many microentrepreneurs lack access to training in basic business skills and, without this support, there is a high risk of failure for microenterprises [57].

COG provided training and practice in basic business skills in a variety of informal ways. The first way was the establishment of small enterprises for the sale of goods such as souvenirs, crafts, cards and snack foods. These skills were further built upon in the operation of the English classes at COG. COG has provided entrepreneurial training for more than four years, encouraging participants to explore various microenterprises before setting on a specific direction. The training provided was informal, as opposed to formal training utilising an accredited curriculum; however, it has been recognised that entrepreneurs often do not engage in formal training due to time and monetary constraints [58]. Furthermore, several recent studies examining business training of microentrepreneurs found no significant positive effects of formal training on business performance, while other studies found business training increased profits, survival or growth in the long or short term [59]. While this current study did not look at long-term findings, it does appear to suggest that hands-on business training provided by COG, not formal training, may provide valid operational knowledge for entrepreneurs. Therefore, the informal entrepreneurship training may have been a factor contributing to the success of the establishment of the microenterprises.

### 4.3.2. Technology Literacy

When establishing microenterprises, it is important to provide quality goods or services that are demand-driven, meeting both current and localised demand. Globally, there has been a rapid change to established economic structures '. . . *agricultural and manufacturing industries have given way to human services as the principal source of economic activity and employment'* [60]. Similarly, in Vietnam, rapid growth has been accompanied by a change in the structure of employment, with agriculture declining and the services sector experiencing growth [61]. This new mode of production offers new employment possibilities for people labelled 'disabled' but does assume access to technology [60].

Technology enables disabled people to participate in a variety of environments [62]. Teachers were able to utilise technology to teach valued and sought-after English classes. Physically disabled teachers interviewed in this study had limitations in hand and arm movements. Whilst this did restrict their ability to write on a black/whiteboard (which is often required when teaching), the teachers adapted their instruction methods. Teachers utilised computers to make PowerPoint presentations and displayed lessons on large television screens instead of using traditional black/whiteboards. Computers were employed to assist physically disabled teachers in preparing lessons, accessing the internet and displaying YouTube clips. This resulted in COG's English classes being unique when compared to most other English classes in Vietnam. As a result, it was reported that parents sent their children to the COG classes because they are highly interactive and more valued than those offered at the local schools.

Studies indicate technology has the potential to boost the competence and autonomy of disabled individuals, emphasising the role of technology in combating the stigma associated with disability [63]. One teacher participant reported she had been fired from her previous preschool position because she was seen as 'diseased' after having a stroke. However, this perceived 'disease' was not a consideration in the COG English classes where technology was utilised. The use of technology can remove both physical and cultural barriers [64]. However, disabled people need opportunities to develop digital technology skills if technology is to lead to empowering disabled people [64]. As a model to be replicated, there is evidence to suggest that the technological adaptations for teachers employed by COG provided adequate skill and knowledge for all to present valued English

classes to at least a primary school standard. This is despite some teachers having limited formal education.

### 4.3.3. Mentors

While entrepreneurship training programs have been found to be ineffective in Vietnam, the use of business mentors has had more success [65]. Director Hoa (Peter) provided mentoring to the disabled teachers for four years and will continue for the teachers as they move out into their own accommodation and establish independent English classes. This support will continue until they are established or if the businesses are not successful.

### 4.4. Empowerment

All the disabled teachers interviewed in this study commented on their lack of confidence before their involvement with COG. Empowerment occurs at several levels, for example, personal empowerment (also referred to as psychological empowerment) and organisational empowerment [66]. Individual empowerment was evident in their new confidence, economic self-sufficiency and belief in the future plans of the teachers interviewed. Future plans included travel, marrying and having a family and opening other microenterprises such as bookstores and fashion shops, all of which the teachers felt confident would happen within a 5–10-year time span.

Organisational empowerment at COG was demonstrated by shared leadership and decision-making. Board members and the Director himself have disabilities, and the Director insists on non-disabled people being offered non-paid positions.

### 4.5. Company of Grace's (COG's) Microenterprise Model

COG's approach to establishing microenterprises integrates elements from various business models, notably drawing from the cooperative model, where individuals with shared interests form a collective [67], and social enterprises designed to address social vulnerabilities [68]. Common advantages inherent in these models are also reflected in COG's model and include the facilitation of skill and knowledge exchange among members and the collaborative use of collective resources to foster a supportive environment for business development. Nevertheless, these models may entail restricted autonomy for members, potentially constraining innovation and agility in decision-making. In contrast, COG's model of providing financial support and training harnesses the strengths of these approaches while simultaneously allowing the entrepreneurs to retain funds earnt, promoting future autonomy and decision-making instead of imposing limitations.

There are several constraints of COG's model. Firstly, it took four years for the disabled teachers to establish their microenterprises, which included learning English, trialling other ventures and learning business skills. Four years may seem a long time to establish a business, especially given microenterprises generally have a high failure rate during their early years [69]. However, a lack of planning has been found to contribute to microenterprise failure [70]. Ensuring teachers are trained in all aspects of self-employment and teaching English is an important step towards ensuring the success of this type of microenterprise. Secondly, start-up capital to purchase premises and technology and provide initial wages was acquired through FBOs. Therefore, this model may not be universally suitable but is highly regarded in this context, especially as obtaining capital prior to planning is also recommended in setting up microenterprises for success [70]. A recommendation for further research is to investigate the long-term sustainability of microenterprises run by disabled people.

Study participants offered instances illustrating how societal attitudinal and structural barriers can adversely affect the employment prospects of disabled individuals, for example, being considered 'diseased' and not being offered writing support or accurate vocational information. In contrast, the COG model tackles systemic challenges to facilitate success in employment for physically disabled people. This is achieved by ensuring appropriate

training, employing adaptations like technology and aligning with the market-driven needs and opportunities within the job market.

### 4.6. Contribution to Knowledge

Disabled people in developing countries are more likely to be self-employed than non-disabled people [6]; hence, this article enhances inclusivity in entrepreneurship literature by filling a gap in existing research, offering further understanding and adding a nuanced perspective to the broader but limited discussion on entrepreneurial development for physically disabled people living in a developing country.

Previous literature has highlighted financial constraints are a common challenge for microenterprises globally [6]. This article presents an alternative to the often-challenging access that disabled individuals face in obtaining loans and business training. Analysing the microenterprise initiative of Company of Grace has highlighted the level of commitment and investment involved in supporting disabled people. This can inform FBOs or other charities and governments on how they can support disabled people in fostering employment outcomes, supporting research from other countries showing that collaboration between government and the private sector may improve the sustainability of microenterprises [71].

This study with its focus on empowerment as a component of entrepreneurial development builds on the work of Neath and Schriner [72] in suggesting the expansion of power, along with strategies in employment, has significant potential to drive social change. COG's redistribution of power within a workplace setting led to an entrepreneurial journey from reliance on others to independence by the study's participants. The COG model offers an approach that not only boosts the confidence of disabled individuals but can lead to broader societal transformations, impacting social dynamics, norms and the promotion of social justice within and beyond the workplace [72].

Despite the presence of policies and services aimed at assisting disabled individuals in Vietnam, it seems that these initiatives are not effectively translating into practical support. This study's analysis of COG's entrepreneurial development, in particular apprenticeship and human resources, addresses the call made by researchers to explore the mediating effects of social capital and entrepreneurial competencies in microenterprise development and sustainability [73,74]. Supporting the existing literature, this study emphasizes the efficacy of infrastructure support, including mentorship [25], and provides an alternative initiative to enhance microenterprise innovation [75].

### 4.7. Limitations

Interview data collected from people involved in COG's microenterprises provided positive impacts, and a larger sample size might have elicited a further range of experiences. Therefore, the findings of this study may not be generalised; however, they contribute to guiding future research endeavours. Furthermore, with private small businesses playing a vital role in Vietnam's economy and the government promoting small- and medium-sized enterprises [71], this study outlines a legitimate employment pathway for physically disabled people. Yet, this model might not be as efficient for individuals with different disabilities. Additional research is necessary to determine which components are applicable to individuals with diverse disabilities.

## 5. Conclusions

This research addresses a gap in the existing literature by examining the establishment of microenterprises catering to physically disabled individuals in Vietnam. These microenterprises, initiated through mentoring, informal training and financial assistance from an Australian faith-based organisation, have successfully created employment opportunities for physically disabled people. Specifically, COG's microenterprise model has empowered individuals to teach English to children, supplementing the children's formal education. Through the implementation of modifications, disabled teachers have conducted interac-

tive classes, enhanced student engagement and introduced a distinctive and high-quality teaching approach uncommon in Vietnamese classrooms. Furthermore, participants in the program have reported heightened confidence, economic self-sufficiency and a positive outlook on their future since joining COG. This model effectively integrates the advantages of various entrepreneurship models while introducing additional strengths. This microenterprise development model might not be feasibly replicated in every situation; it does, however, make a significant contribution.

**Author Contributions:** Conceptualization, J.A.; methodology, J.A.; formal analysis, J.A.; resources, J.A.; writing—original draft preparation, J.A.; writing—review and editing, C.H. and G.C. All authors have read and agreed to the published version of the manuscript.

**Funding:** This research received no external funding.

**Institutional Review Board Statement:** The study was conducted in accordance with the Declaration of Helsinki, and approved by Flinders Human Ethics Committee (reference: 7290).

**Informed Consent Statement:** Informed consent was obtained from all subjects involved in the study.

**Data Availability Statement:** Participant consent did not extend to data sharing with third parties.

**Acknowledgments:** The authors would like to sincerely thank the participants of this study for their contribution and wish microentrepreneurs every success in the next development of their businesses. In memory of Hoa's valuable contribution to disabled people in Vietnam. Hoa (Peter) Van Stone, 27 February 1967–12 April 2021. Further information about Hoa's life can be found in his autobiography: *Heart of Stone: My Story*. Finsbury Green Printing. The authors thank Amy White for contributing to this article and Thanh Ha Nguyen for translation services.

**Conflicts of Interest:** The authors declare no conflicts of interest.

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
