# Peer review of "Empowering Physically Disabled People in Vietnam: A Successful Microenterprise Model"

_disabilities, doi:10.3390/disabilities4010009_

Round 1

Reviewer 1 Report

Comments and Suggestions for Authors

This paper claims how a microenterprise could empower their employees with physical disability in Vietnam. The topic is little understood and little studied. The passion of the author for the topic is evident and they are to be commended for taking on this research.  However, it would require considerable revision before it could be recommended for publication. I strongly encourage the author to take measures to bring the paper into better alignment with the standards of quality published in "Disabilities". The comments below are meant to provide specific guidance and in no way should be taken as discouragement:

1.       Abstract: The author claims that the director supports six physically disabled entrepreneurs, but they are employees in the COG, not entrepreneurs. The author must clarify or change the term.

2.       The author has not critically and thoroughly reviewed published evidence regarding the strengths and challenges of disabled employees. Additionally, this paper focuses solely on physically disabled employees without providing a specific rationale.

3.       Many key terms (e.g., microenterprise, self-employment, mild form of disability, etc.) are used throughout this paper without clear definitions and theoretical foundations. Moreover, the author should clearly justify their theoretical stance on disability (p.2) for this paper, as it is crucial to establish the identity of disabled employees.

4.       The author needs to provide statistical information on the number of disabled people and the number of disabled people in employment in Vietnam in P.2. This would be helpful for readers to understand the study's context.

5.       On page 2, did the Vietnam government introduce any legislation for a disabled employment quota system? If so, the author must critically examine them.

6.       On page 3, the author must provide information on how many vocational training schools for disabled people exist in each sector (i.e., government, NGOs, and private organizations). The author claims that 60% of training participants get a job after training, but it is unclear what kind of employment they secure and whether it's part-time or full-time after training.

7.       What physical and attitudinal barriers have disabled people experienced when accessing job placement and counseling centers? Why?

8.       How many faith-based organizations for disabled people are there in Vietnam, and what role do they play in the country?

9.       On page 4, the author claims that this study investigates the approach and outcomes of COG using Lingelback et al. (2005)'s three attributes of entrepreneurship. However, it's unclear why they use this model as an analytical lens. This paper is about the formalization of a microenterprise, not the effects of an entrepreneurship program. The author may need to reconsider their analytical framework and focus for this paper.

10.   On page 5, there is confusion regarding the number of interviews conducted (line 204 vs. line 206). This needs to be clarified.

11.   It would be beneficial for readers if the author could provide a table detailing the characteristics of the study participants.

12.   On page 5, line 217, how many participants reported having a disability from birth or a young age?

13.   On page 5, line 235, it is not clear how the author developed the semi-structured interview for data collection.

14.   On page 5, Line 240, I wonder whether the author could gather enough high-quality data within 15- or 36-minute interviews with disabled employee participants.

15.   On page 6, the author claims that data are analysed using content analysis, but there is no explanation of the process of content analysis or how the two different analysis approaches (i.e., content analysis and theory-guided codes) were merged.

16.   There are interesting findings, but the author has summarised the findings from each participant narratively. Therefore, this section needs to be rewritten to highlight the major contradictory, similar, or unique findings from the interview data.

17.   I wonder why COG did not access government funds to establish a microenterprise instead of relying on an Australian FBO's donation.

18.   The discussion section should engage in a critical dialogue between this study and the existing literature. However, the current version of this section appears to be another findings section because it introduces many new pieces of information that were not introduced in the findings section.  Importantly, this paper needs to engage in a critical discussion comparing the current microenterprise business models with COG's business model.

19.   The author claims that COG provides support for disabled employees to start their own businesses. However, in this paper, I have found that all the support from COG is informal. Were there any formal entrepreneurial training programs for disabled employees in COG? There is no empirical evidence in this paper regarding the impact of such informal learning on improving disabled employees' entrepreneurial skills or employment outcomes.

20.   On page 12, the conclusion section is very weak. It appears that the author is repeating themselves.

Comments on the Quality of English Language

There are numerous repetitions throughout this paper. The author must revise it to eliminate unnecessary repetitions.

Author Response

We attach a document that details our response to the feedback of both reviewers. We thank them for their helpful comments in revising this manuscript. 

Reviewer 2 Report

Comments and Suggestions for Authors

This paper provides a good description of the situation people with disabilities face in Vietnam, as well as documenting a small, but promising model for microenterprise development for people with disabilities.  But this is a study about people with PHYSICAAL disabilities and microenterprise. That should be made clear in the title and description of the research right from the start. People with intellectual and psychosocial disabilities or hearing or vision difficulties face a different set of barriers, and may experience these programs differently. I would have like to have seen the authors zoom out a bit and explain how such a model could adapt to provide opportunities for other people with disabilities. Deaf people aren't going to teach English. But are there lessons learned on how COG developed its program that could be applied to people with other types of disabilities?

Also, Decree No 80/2021/ND-CP (Decree 80) issued by the Vietnamese government guides the implementation of the Law on Support for Small and Medium Enterprises (SMEs). In fact, SME's account for 40% of GDP and 50% of employment. Why has the SME program not worked for people with disabilities? It would be great if they could've reported some specific info on why the SME program is not working for people with disabilities. There was a lot of good documentation of discrimination and barriers in general, but it would be great if problems with the SME program, in particular, could have been addressed. Is it possible to do this now?

Finally, a minor point: In my experience the term Nguoi Tan Tat is used a lot less frequently now, as opposed to Nguoi Khuyet Tat which is preferred by people with disabilities in Vietnam. I think I would leave that out, as Vietnamese might see that as a sign that the discrimination material is outdated -- and I don't think it is, so I wouldn't want to leave that impression.

Author Response

(The authors gave the same response as above.)

Round 2

Reviewer 1 Report

Comments and Suggestions for Authors

I have found that the authors addressed many issues in the current revised version. However, I have identified a few areas that need further clarification:

  • On Line 32, the authors claim that they adopted the biopsychosocial model of disability for this paper. However, it is not entirely clear how this model was applied in this context.

  • On Line 118, the authors need to provide an example of a home-based business.

  • On Line 126, it is unclear what the statement 'The Vietnamese attitude of care means people with disabilities often received food' means.

Author Response

The file will not attach so please find responses below:

  • On Line 32, the authors claim that they adopted the biopsychosocial model of disability for this paper. However, it is not entirely clear how this model was applied in this context.

Response: 

Adopting this model offers a thorough comprehension of factors such as societal attitudes, resource accessibility, the impact of disability on productivity, and how these elements shape the experiences and outcomes of individuals with disabilities in the context of microenterprises. Lines 33-36

  • On Line 118, the authors need to provide an example of a home-based business.

Response: 

A home-based business is an enterprise or commercial venture that is operated and managed from a residential location, typically the business owner's home. Home-based businesses are favored by individuals with disabilities, potentially due to challenges encountered when seeking traditional employment opportunities [32]. Lines 120-124

  • On Line 126, it is unclear what the statement 'The Vietnamese attitude of care means people with disabilities often received food' means.

Response: 

The Vietnamese cultural emphasis on care and protection results in individuals with disabilities receiving essential care, food, and shelter. However, societal inclusion, particularly in workplaces, remains limited due to the prevalent belief that people with disabilities are perceived as incapable. Lines 131-135
